# In the Opinion of Elite Volleyball Coaches, How Do Contextual Variables Influence Individual Volleyball Performance in Competitions?

**DOI:** 10.3390/sports10100156

**Published:** 2022-10-18

**Authors:** Carlos López-Serrano, María Perla Moreno Arroyo, Daniel Mon-López, Juan José Molina Martín

**Affiliations:** 1Departamento de Deportes, Facultad de Ciencias de la Actividad Física y del Deporte – INEF, Universidad Politécnica de Madrid, 28040 Madrid, Spain; 2Departamento de Educación Física y Deportiva, Facultad de Ciencias del Deporte, Universidad de Granada, 18071 Granada, Spain

**Keywords:** coaching, performance analysis, sport, competitive environment, practical perspective

## Abstract

The main objective was to know the elite coaches’ opinions regarding the relevance, definition, and importance of volleyball contextual variables to measure individual performance in competition. After performing a literature review, an instrument to gather the opinion of the world’s elite volleyball coaches was elaborated by four volleyball specialists. The sample of experts consisted in 20 world’s elite volleyball coaches who met at least three experience years in first division or national teams. The instrument collected experts’ information on the contextual variables in relation to relevance, definition, and importance. Cronbach’s α and Aiken’s V coefficient were used to test the reliability and content validity of the contextual variables, respectively. To compare the importance of the contextual variables U de Mann-Whitney and Kruskal-Wallis tests were used. Results showed that opposition level, set period, score difference, results of the previous set, competitive load variables, high level, final periods ≥20, and high load categories were relevant (Aikens V > 0.70). In addition, high level, final period ≥20 and ≥+10, and high load categories were significantly more important (*p* < 0.05). We conclude that, according to the elite coaches, the contextual variables should not be analyzed separately. Future studies should consider contextual variables dynamically.

## 1. Introduction

Professional coaches usually measure players’ performance to predict their behaviors, as coaches’ decisions might have some kind of influence on team wins and losses [1]. Thus, the traditional intuition of coaches is being progressively replaced by the collection of accurate data to find the ‘Win share’, which is an estimation of individual player performance related to their team’s wins [2]. This analysis would facilitate the focusing of players’ attention on those aspects that coaches consider most important to improve their performance [3].

The performance of team sports depends on multiple factors. Several studies have analyzed performance from different fields using physical, technical-tactical, psychological, or even contextual variables [4]. Moreover, it is vital to identify players’ performance within ecological contexts [5]. Competitiveness in elite sports has increased; thus, any information to improve player performance could provide an advantage over the opponent [6]. However, there seems to be a mismatch between the scientific field and practice because coaches are reticent to give information and many studies are under controlled environments outside competitive situations [7]. Consequently, a holistic perspective of competitive performance requires investigations in ecological contexts alongside coach collaboration [8].

In recent years, technological advances, such as artificial intelligence, algorithms, and big data, have improved the ability to collect data in game contexts, involving coaches and researchers [9]. In this line, data have grown exponentially in recent years, and sports technology has allowed their analysis. This fact has involved the creation of a new work area, namely Sports Analytics, and the birth of a new professional figure, the expert observer within the coaching staff, called a scout or data analyst [3].

Specifically, in volleyball, the data recording is carried out using the Data Volley software [10], a system by federations, clubs, and coaches to evaluate performance. These data are often used to show coefficients and descriptive statistics (e.g., averages, efficiencies) [11], focused on recording and quantifying the technical actions of players in contact with the ball [10]. While some contextual variables can be registered with data volleyball (e.g., home advantage), these variables do not have a standardized approach, and their analysis often lacks common criteria for all coaches [12]. However, this weakness in the analysis of contextual variables contrasts with their influence on sports performance [13].

The moment of the match or the scoreboard seems to affect the players’ behaviors, thereby disrupting the performance values of technical actions [14]. Additionally, these contextual variables have been defined as different situations or external competitive factors that can affect the performance of both individual players and the wider team [15]. In this line, previous literature has shown that variables like home advantage [16], the difference between opposite team levels [17], or the final moments of the set and the match [18] have an influence on the final result. Consequently, contextual variables such as home advantage, opponent level, set period, score differences, trend, the result of the previous set, competitive load, etc., could be critical in volleyball performances.

In particular, volleyball is conditioned to Quasi-Simpson’s paradox scoring system [19]. This paradox implies that the winning team is not always the team that gets the most points during the match. Interestingly, although all the points have the same quantitative value, coaches and researchers have confirmed the existence of situations, critical moments, or “momentum” with a greater impact on the match result [20]. This term has been used to describe strong feelings that generate confidence for an outcome which are related to successful previous actions [20]. Determining those performance indicators could facilitate coaches’ decision-making and, as a result, later wins [7,21]. However, considering this previous information, what indicators should be measured? Additionally, what contextual variables are the most important for the coaches?

Although previous research has highlighted the importance of contextual variables and their influence on performance [13,17], the exclusive categorization of technical actions in volleyball could be a limitation in player analysis. In addition, there seems to be limited practical evidence supported by expert coach opinions on the validity and consistency of such methods to analyze the influence of contextual variables, even though science has considered it essential to integrate elite coaches’ practical knowledge of sports performance [22]. Consequently, the main objective of this study was to identify the opinions of elite coaches, in terms of relevance, definition, and importance, on contextual volleyball variables to measure individual performance in competitions.

## 2. Materials and Methods

### 2.1. Instruments

An instrument to gather the opinion of the world’s elite volleyball coaches was elaborated upon by four volleyball specialists. The specialists in question were university professors specializing in volleyball with at least five years of experience. The instrument was designed in two steps: (I) a bibliographic review of the most frequently studied contextual variables in sports research was carried out (see Table 1); and (II) a bibliographic review of the most commonly studied contextual variables in volleyball was carried out, which led to the development of the instrument (see Table 2).

### 2.2. Participants

To obtain the opinion of the world’s elite volleyball coaches, the instrument was sent to a group of expert judges. Only those expert judges who possessed at least three years of experience in one of the following criteria were considered international elite coaches and included in the study: (I) national team coach; (II) first division team coach in their respective country; or (III) coach in international competitions. Initially, 40 expert judges were contacted but only 20 of them participated in the study (50% response rate). Following the classification pointed out by Swann et al. [117], the final characteristics revealed that the sample of international elite coaches included eight elite competitive, eight elite successful, and four elite world-class coaches across three continents. Coaches have served in the first divisions of Switzerland, France, Colombia, Peru, Spain, Greece, Puerto Rico, and Belgium, highlighting the Italian, Brazilian, or Turkish leagues and national teams in Colombia, USA, Korea, Belgium, Spain, or Canada. Additionally, they have competed in the Olympic Games, the European Volleyball Confederation Champions League, and the South American Volleyball Championship. Lastly, regarding the teams’ gender distribution, the experts were divided into women (70 %; *n* = 14) and men team coaches (30 %; *n* = 6). All participation in this study was voluntary, and participants signed an informed consent form before data were collected.

### 2.3. Procedures

The instrument aimed to analyze the contextual variables and their categories.

The contextual variables were analyzed by two criteria: (I) Relevance: adequacy and suitability level of the variable to measure the individual match performance of each player; and (II) Definition: level of clarity of the variable denomination and definition. In addition, the categories were analyzed using relevance, definition, and important variables. Importance was defined as the category value to measure individual performance.

A scale from one to ten, in one-point intervals, was used to measure definition, relevance, and importance, with one being “totally disagree” and ten “totally agree”. Moreover, at the end of each section, the experts were asked to write their suggestions or comments. A SurveyMonkey online platform was used to host the instrument. An online link was sent directly to each world elite coach containing information about the procedure and objectives of the study. The instrument was available for one month to facilitate the participation of experts alongside their numerous commitments to international competitions. Several follow-ups were sent out seven and three days before the platform was closed. No more answers were accepted after the instrument deadline.

### 2.4. Data Analysis

The reliability of the contextual variables and their categories was checked using Cronbach’s α and McDonald’s Omega. The reliability in our study was: home advantage α = 0.937, ω = 0.937; opposition level α = 0.964, ω = 0.966; set period α = 0.887, ω = 0.892; score difference α = 0.813, ω = 0.844; trend α = 0.957, ω = 0.959; result previous set α = 0.963, ω = 0.963; and competitive load α = 0.741, ω = 0. 746. To determine the content validity of contextual variables and their categories, Aiken’s V coefficient was calculated. Confidence intervals at 90%, 95%, and 99% were estimated using the visual basic program for the Aiken V developed by Merino & Livia [118] through the scoring method [119]. To compare the quantitative differences assigned by the coaches to the importance of the contextual variables with two categories (home advantage, result previously set, and competitive load), the Mann-Whitney U test was utilized, while the Kruskal-Wallis test was used in those variables with three or more categories (opposition level; set period; score difference and trend). The mean (M), median (Md), and standard deviation (SD) were used to describe the variable’s importance. The normality of the variables was verified using the Shapiro-Wilk test; the significance level was set at *p* < 0.05. The data were analyzed using the SPSS v.25 statistical package (IBM Corp., Armank, NY, USA).

## 3. Results

The Aiken’s V coefficient values of the contextual variables, their categories, and the confidence intervals, according to their relevance, definition, and importance, are illustrated in Table 3.

All the variables and categories achieved validity values over 0.50 [120]. However, not all the variables and categories obtained validity levels that could be considered adequate (0.70) [121]. Consequently, those variables and categories with values of 0.67 were reviewed according to the qualitative assessment of the experts.

### 3.1. Relevance of the Contextual Variables

The following variables and categories showed validity values higher than 0.70 (see Table 3): variable II—opposition level; high-level category; variable III—set period; final period category (≥20) 1st—4th set; final period category (≥10) 5th set; variable IV—score difference; variable VII-competitive load; and high competitive load category. The remaining variables and categories presented values of 0.67, with the following main comments by the expert judges:-Variable I—Home advantage: “is more important to prevent and work, than to evaluate” (Expert Judge 11).

Home advantage categories: “It can be different depending on the moment of the championship” (Expert Judge 15).

-Variable II—Opposition level: “pre-set[ting] the opposition levels may lead [to the loss of] a lot of information” (Expert Judge 3).

Opposition level categories: “Pre-set[ting] the level ranges generally and symmetrically can be a problem” (Expert Judge 3); “The competitive performance has to be adjusted and measured in terms of similar levels” (Expert Judge 12); “The championship moment and the team’s psychological status can affect the opposition level” (Expert Judge 15).

-Variable III—Set period: “Analyzing the set period [but] not considering the scoreboard difference is a big error. It’s not the same scores of 24–23 or 24–17. Set period variable and score difference should be analyzed together” (Expert Judge 4).

Period set categories: “This variable is related to the scoreboard, favorable or against” (Expert Judge 15); “The final periods are very important, but only with tight scoreboards” (Expert Judge 16); “The point differences between both teams should be [considered], 23–11 is not the same as 13–23. Maybe the 5th set could be divided into two periods only (0–8/9…)” (Expert Judge 17).

-Variable IV—Score difference: “This variable should be related to the set period” (Expert Judge 11); “score difference [has] a great relationship with the variable III-set period” (Expert Judge 13).

Score difference categories: “The established intervals are not adequate, since the intervals should have a dynamic amplitude depending on the category, the set period, or the scoreboard” (Expert Judge 3); “To improve the relevance, the variable could be related to the set period, especially from point 16 and from point 20 onwards” (Expert Judge 11); “According to its globality, this variable is critical in the 5th set” (Expert Judge 12).

-Variable V—Trend: “The score difference and trend variables could be mixed. To evaluate the team’s actions, it is not the same to have scores of +2 or −2, +4, or −4, as the teams will play differently. Therefore, I think that score difference and trend should be the same variable” (Expert Judge 4); “Competing in intervals of ±2 points is like going tied. [The] trend would be [of] more relevance if it were related to the set period” (Expert Judge 11).

Trend categories: “Only low scoreboard differences would have [an] influence” (Expert judge 13).

-Variable VI—Result previous set: “The championship round should be added” (Expert Judge 15).-Variable VII—Competitive load: “you should add the championship round” (Expert judge 15).

### 3.2. Definition of the Contextual Variables

All the variables and categories showed validity values higher than 0.70, with the exception of the following variables and categories, which obtained values of 0.67 (see Table 3): away category (of variable I—home advantage); variable II—opposition level; low level+; initial period (0–4) 5th set (of variable III—set period); high differences (+5) 5th set (of variable IV—score difference); and variable V—trend. Accordingly, the main comments from the expert judges were as follows:

Home advantage categories: “Please include the neutral field category” (Expert judge 2); “the definition of the home advantage categories must not consider the public influence, since you can play at home or away, without spectators in the stands (i.e., season 20–21)” (Expert judge 4).

Opposition level categories: “It is difficult to understand conceptually” (Expert Judge 11).

Set period categories: “Maintaining the same number of intervals and a proportional distribution of points in the 5th set [complicates] the possibility of analyzing comparisons between the 5th and the rest of the sets. I proposed that the final period should be maintained (last 5 points)” (Expert Judge 3).

Score difference categories: “In the 5th set, I would consider; low differences 0 and 1, mean differences 2–3, high differences 4 or more score difference points” (Expert judge 4).

### 3.3. Importance of the Contextual Variables

The categories with a content validity of higher than 0.70 were (see Table 3): high level (the variable II—opposition level); final period (≥20) 1st—4th set and final period (≥10) 5th set (variable III-set period); and high (the variable VII—competitive load). The remaining categories presented values of 0.67.

Additionally, a quantitative analysis was conducted to check the experts’ importance values in all categories; these values are described in Table 4. Results demonstrated differences between the Set Period categories (X^2^(5) = 18.6; *p* = 0.002). The Bonferroni post hoc analysis also showed the following differences between periods: final period ≥20 has higher values than 0–9 (*p* = 0.003), 10–19 (*p* = 0.027), 0–4 (*p* = 0.012), and 5–9 (*p* = 0.026). The final period ≥10 has higher values than 0–9 (*p* = 0.001), 10–19 (*p* = 0.017), 0–4 (*p* = 0.007), and 5–9 (*p* = 0.016). In addition, high load categories had higher values that attenuated the load (Z = 2.06; *p* = 0.040). For the rest of the comparisons, no differences were found (*p* > 0.05).

## 4. Discussion

The main objective of this study was to identify the opinions of elite coaches, in terms of relevance, definition, and importance, on contextual volleyball variables to measure individual performance in competitions. For this purpose, a valid and reliable instrument was created.

The main results of this study ascertain that there are differences between the contextual volleyball variables previously analyzed in the literature. The opposition level, set period, score difference, result of the previous set, and competitive load variables were considered to be relevant by experts in measuring individual player performance in competitive environments. Conversely, the home advantage and trend variables did not achieve an optimal level of relevance.

Neither the home advantage variable nor the home and away categories was considered to be relevant by experts. In fact, expert coaches suggest that it is more important to train than to evaluate the effect of the home advantage. Similarly, Palao et al. [16] suggested that psychologically preparing athletes for the noise level of the audience, as well as familiarizing players with the playing court the day before could be necessary for maintaining their performance level. However, the effect of home advantage has been repeatedly studied in sports [34], specifically in volleyball, with inconclusive results [16]. Thus, the factors that seem to determine the influence of home advantage in volleyball are the crowd, familiarization with the playing court, disruptions to sleeping or eating habits caused by traveling, and the referees [30]. Subsequently, there seems to be a difference between the relevance of home advantages in our study and the previous literature. This could be due to the fact that coaches are able to counteract the effects of home advantage with training; additionally, home advantage has an enormous dependence on other contextual variables such as set period, score difference, and opponent level [37].

Regarding the definition of home advantage, experts suggest the inclusion of a neutral field category, as seen in single-court competitions such as the Olympic Games. Moreover, experts do not consider it appropriate to include public influence in the definition of the home advantage variable, as some matches have no audience. In this regard, a recent study by Correia-Oliveira and Andrade-Souza [34] indicated a decrease in the home advantage effect in post-pandemic situations with empty stands. In addition, our results failed to discover any significant differences concerning the importance of home or away categories. However, our data reflect slightly higher values for away teams in contrast to home teams. Similarly, Yu et al. [13] quantified the probability of winning as 57% for the home team.

The results showed that the opponent level variable and the high-level category (matches between teams at similar levels) were considered to be relevant by the experts, in accordance with previous literature. Thus, the opponent level determines how teams play and their performance [11]. Furthermore, the elite coaches indicated that pre-defining the opposition level according to the competition ranking could involve the loss of a significant amount of information. Similarly, Marcelino et al. [11] proposed the use of probability functions of the winning numbers, the won/lost ratio of the points, the set win/loss ratio, and the percentage of sets won in contrast to the final ranking. These indicators could give better information about performance fluctuations throughout the competition. Lastly, our results revealed significant differences in the importance of the high-level categories (9.5/10). Accordingly, the best player performances occur in highly competitive or balanced matches, where team performance levels are similar [17].

The set period variable, the final period ≥20, and the final period ≥10 set categories were considered to be relevant by the experts. However, they suggested that the set period should take into account the score difference variable. The previous literature has highlighted the greater impact of score differences compared to the set period. Specifically, in volleyball, points above 20 in the 1st to 4th sets with closed scoreboards seem to be critical [64]. This suggests that the impact of the final set moments (final period ≥ 20 and final period ≥ 10) would be conditioned by tight scoreboards. Thus, player performances could be particularly affected by stress or psychological pressure in the final periods [18].

In the present study, the set period variable was defined using categories of (0–9), (10–19), (20+) for the 1st to 4th sets and (0–4), (5–9), and (10+) for the 5th set, according to García-de-Alcaraz and Usero [61]. Contrary to our definition, one expert suggested setting the categories according to the technical time-out presented in the 2016 International Volleyball Federation (FIVB) rules—points 8, 16, and 21 for sets 1 to 4 and only two categories for the 5th set before and after the 8th point. Similarly, Rodriguez-Ruiz et al. [65] established three periods: (0–8), (9–16), and (16+) points. However, our category proposal is based on the suppression of the technical time-out by the present FIVB rules 21–24 [122]. Regarding the importance of the set period categories, the final periods ≥20 and ≥10 achieved significantly higher values than the initial or central period of all sets. Both periods (final period ≥20 and final period ≥10) received the maximum possible values (10/10) in contrast to the remaining periods. The results are in line with those of Marcelino et al. [18], which suggested the crucial role of the final set period in the competition result.

The results determined that the score difference variable was relevant but not the category thereof. The final periods of sets with score differences of less than two points (“closed sets”) seem to be determinant [75]. Moreover, the relationship between the score difference and set period variables has been highlighted previously; the relevance of the set according to the scoreboard has also been classified into ambivalent sets (with 0–2-points difference), safe sets (3–5-point difference), and unbalanced sets (with advantages greater than 5 points) [55]. Thus, similarly to our study, the previous literature has defined ranges of 0–2 points as low differences [55]. Along this line, all the experts except one (who pointed out the need to include close point ranges (0 or 1 for the 5th set) agreed with the definition of score differences in our study. On the other hand, we did not find significant differences in the score difference categories. However, in accordance with Dávila-Romero and García-Hermoso [75], who showed the greater victory influence of “closed sets”, we discovered that the maximum value was achieved for the low differences category, specifically 0–2 points for the 1st to 4th sets.

Neither the trend variable nor its categories were considered to be relevant by experts. In addition, the experts suggested that it would be more relevant if the trend was linked to scoring difference and set period. Accordingly, Jeon and Park [84] considered performance trends as a consequence of random processes associated with each action. In contrast, Stanimirovic and Hanrahan [78] found that positive trends in the scoreboard or “momentum” were associated with psychological effects that affect performance. Similarly, sports like tennis with Simpson’s paradox scoring system [19] have shown a positive effect as a result of winning the previous point [86].

Regarding the definition of trend, one expert pointed out that score difference and trend should be the same variable. In this topic, the trend variable has been called “score-line” and the previous literature suggests that this variable affects performance in the final periods of closed or tied matches [79]. Moreover, unbalanced scores could decrease players’ motivation and reduce their efforts [59]. Consequently, considering both variables together could be necessary.

No significant differences were revealed as important between the trend categories. Additionally, one expert noted that trend categories would have an influence only when this variable is analyzed with low differences in the score together. Although there seems to be a lack of knowledge in volleyball, Sampaio et al. [59] established that the score-line is an important factor in measuring basketball performance since winning teams can lose concentration because of the sense of accomplishment.

The result of the previous set variable was considered to be relevant by the experts, but not its category (won or lost). Nonetheless, one expert did not believe in the existence of a direct correlation between the results of the previous set and the set-in play. In the absence of specific studies in volleyball, Dietl and Nesseler [123] showed the existence of a carryover effect between the previous and subsequent sets in tennis. This carryover effect could be related to psychological factors and the existence of “momentum”. Regarding the definition of the result of the previous set, one expert proposed including the championship round. In this line, Raymond et al. [64] suggested that losing the previous set would add psychological pressure in advanced rounds of knockout competitions, and the championship round could be an additional factor that should be taken into account. On the other hand, no differences were found between the importance of the categories (Lost —8 vs. won—7). A possible explanation of this result could be that winning teams tend to relax, in contrast to the losing teams who increase their efforts to improve their performance [14]. Nevertheless, Dietl and Nesseler [123] showed that winning the previous set could stimulate winning the next set, confirming the carryover effect in tennis.

Both the competitive load and high load category variables were deemed to be relevant by experts to the measurement of individual performance. As expected, the previous literature has related this relevance to psychological pressure in critical situations such as close matches or nearing the end of a match [20]. Moreover, divisions such as non-decisive and decisive sets have also been proposed previously by Ramos et al. [101]. On the other hand, experts suggest adding the championship round to the definition. Similarly, several sports have paid attention to competitions with elimination rounds [43], as there is an increase in pressure during qualifying matches of the final championship rounds [64]. Finally, the high-level category was considered to be significantly more important by the experts (10/10 high load category versus 7/10 category attenuated load). This may be due to players tending to decrease their performance when they feel there is little chance of them winning and thus trying to save energy to more effectively recover for the next set [19].

Although this study is one of the first to analyze the influence of contextual variables from the point of view of elite coaches, some limitations should be mentioned, including the study investigating only the elite level and not having analyzed gender differences. Lastly, as a practical application, this study can help the literature to better understand how contextual variables influence individual volleyball performances. This knowledge could be useful for coaches as volleyball training sessions could be modified according to the contextual variables influencing specific training situations that facilitate the improvement of players’ performance in competitions. In addition, this information could be the baseline upon which to elaborate future algorithms or performance coefficients.

## 5. Conclusions

According to our results, we can conclude that only the opposition level, set period, score difference, result of the previous set, and competitive load variables were deemed to be relevant by the experts to measure individual player performances in competitions. Moreover, it would be advisable to consider the championship round within the contextual variables like the result of the previous set or competitive load. Similarly, categorizing the opposition level variable dynamically and not through the team’s final classification would avoid the loss of relevant information.

The different game contexts could induce the appearance of “momentum” or critical situations, with high psychological pressure components on the players. Consequently, the high level (opposition level variable), final period of all sets (set period variable), and high load (competitive load variable) categories could be determinants in the performance as these situations are related to stressful situations with minimum possibilities of recovery. Finally, the elite coaches suggested that the performance or contextual variable analysis should not be done in isolation. Consequently, the dynamic nature of the game and its contextual variables imply that a complete and comprehensive analysis is necessary to measure individual players’ performance in competitions.

## Figures and Tables

**Table 1 sports-10-00156-t001:** Review of the most studied contextual variables in sports research.

Name of the Variables	Sports	Authors
Home advantage Match Location	Baseball	[23]
	Basketball	[24,25,26,27,28,29,30,31,32]
	Football	[33,34,35,36,37]
	NHL, Ice Hockey	[38,39]
	Several Sports	[40,41,42,43]
	Volleyball	[13,16,44,45]
Opponent level Quality of Opposition	Basketball	[32,46]
	Football	[36,47,48,49,50,51,52,53]
	Hockey	[38]
	Volleyball	[4,11,17,18,54,55,56]
Game period Set period	Basketball	[46,57,58,59]
	Tennis	[60]
	Volleyball	[17,18,55,61,62,63,64,65]
Score differences Closed matches Scoring Rates Balance/unbalance perception	Basketball	[25,28,31,32,66,67,68]
	Football	[35,69,70,71]
	Tennis	[72,73,74]
	Volleyball	[4,11,56,61,62,63,64,65,75,76,77]
Scoring rhythm Score-line Trend Match Status Game Status Current on the scoreboard	Basketball	[59,78,79]
	Football	[14,70,71,80,81]
	Hockey	[38,82]
	Several Sports	[83,84,85]
	Tennis	[86]
	Volleyball	[11,61,62,87,88]
Previous period results Previous set result Previous action result	Basketball	[89]
	Football	[14]
	Tennis	[19,60,74,90,91,92,93]
	Several Sports	[83,94]
	Volleyball	[64,87,88]
Competitive Load	Basketball	[27,95]
	Several Sports	[20,43,96,97,98]
	Tennis	[19,60,99]
	Volleyball	[11,62,63,76,100,101]
Scoring First	Football	[50,102]
	Hockey	[38]
	Tennis	[19,92,99]
Match Congestion	Basketball	[32]
	Tennis	[103]
Type of a match	Basketball	[104,105]
	Football	[106,107]
	Volleyball	[61]
Rotation Tactical systems Game models Tactical principles	Basketball	[69,108]
	Football	[109,110,111]
	Several Sports	[112]
	Volleyball	[56,62,63,113,114,115,116]

**Table 2 sports-10-00156-t002:** Instrument with the most studied volleyball contextual variables and their categories in volleyball.

Variable	Categories		Description
I. Home Advantage (HA)	Home		Home team
	Away		Away team
II. Opposition Level (OL)	Low Level−		2 groups above 2 groups below 1 group above 1 group below Same level
	Low Level+		
	Mid-Level−		
	Mid-Level+		
	High Level		
III. Set Period (SP)	1st–4th set	5th set	
	Initial Period (0–9)	Initial Period (0–4)	Initial Period (0–9): from 0 to 9 points in (1st–4th set) Initial Period (0–4): from 0 to 4 points in (5th set)
	Central Period (10–19)	Central Period (5–9)	Central Period (10–19): from 10 to 19 points in (1st–4th set) Central Period (5–9): from 5 to 9 points in (5th set)
	Final Period (≥20)	Final Period (≥10)	Final Period ≥ 20): ≥20 points (1st–4th set) Final Period ≥ 10): ≥10 points (5th set)
IV. Score Difference (SD)	1st–4th set	5th set	
	Low 0–2	Low 0–2-(5th)	Low difference, between 0–2 points (1st–4th set) Low difference, between 0–2 points (5th set)
	Mean 3–5	Mean 3–5-(5th)	Mean difference, between 3–5 points (1st–4th set) Mean difference, between 3–5 points (5th set)
	High +5	High +5-(5th)	High difference, +5 points (1st–4th set) High difference, +5 points (5th set)
V. Trend (TR)	Winning		team winning on the scoreboard
	Losing		team losing on the scoreboard
	Tied		both teams tied on the scoreboard
VI. Result of the Previous Set (SETp)	Won		win previous set
	Lost		lost previous set
VII. Competitive Load (CL)	Attenuated Load		1st, 2nd (3rd sets in case of a set tie)
	High Load		4th & 5th set (3rd set if enabled to win)

**Table 3 sports-10-00156-t003:** Relevance, definition, and importance Aiken’s V values of the contextual variables and their respective categories.

			Interval Confidence Values							Interval Confidence Values							Interval Confidence Values					
		Aiken’s V	90%		95%		99%		Aiken’s V	90%		95%		99%		Aiken’s V	90%		95%		99%	
		REL	Inf	Sup	Inf	Sup	Inf	Sup	DEF	Inf	Sup	Inf	Sup	Inf	Sup	Imp	IMP	Sup	Inf	Sup	Inf	Sup
Variable I	Home advantage (HA)	0.67	0.61	0.72	0.59	0.73	0.57	0.75	0.78	0.72	0.82	0.71	0.83	0.69	0.85	-	-	-	-	-	-	-
Categories	Home	0.67	0.61	0.72	0.59	0.73	0.57	0.75	0.78	0.72	0.82	0.71	0.83	0.69	0.85	0.67	0.61	0.72	0.59	0.73	0.57	0.75
	Away	0.67	0.61	0.72	0.59	0.73	0.57	0.75	0.67	0.61	0.72	0.59	0.73	0.57	0.75	0.67	0.61	0.72	0.59	0.73	0.57	0.75
Variable II	Opposition Level (OL)	0.78	0.72	0.82	0.71	0.83	0.69	0.85	0.67	0.61	0.72	0.59	0.73	0.57	0.75	-	-	-	-	-	-	-
Categories	Low Level−	0.67	0.61	0.72	0.59	0.73	0.57	0.75	0.78	0.72	0.82	0.71	0.83	0.69	0.85	0.67	0.61	0.72	0.59	0.73	0.57	0.75
	Low Level+	0.67	0.61	0.72	0.59	0.73	0.57	0.75	0.67	0.61	0.72	0.59	0.73	0.57	0.75	0.67	0.61	0.72	0.59	0.73	0.57	0.75
	Mid Level−	0.67	0.61	0.72	0.59	0.73	0.57	0.75	0.78	0.72	0.82	0.71	0.83	0.69	0.85	0.67	0.61	0.72	0.59	0.73	0.57	0.75
	Low Level+	0.67	0.61	0.72	0.59	0.73	0.57	0.75	0.78	0.72	0.82	0.71	0.83	0.69	0.85	0.67	0.61	0.72	0.59	0.73	0.57	0.75
	High Level	0.78	0.72	0.82	0.71	0.83	0.69	0.85	0.78	0.72	0.82	0.71	0.83	0.69	0.85	0.78	0.72	0.82	0.71	0.83	0.69	0.85
Variable III	Set Period (SP)	0.78	0.72	0.82	0.71	0.83	0.69	0.85	0.78	0.72	0.82	0.71	0.83	0.69	0.85	-	-	-	-	-	-	-
Categories	Initial Period (0–9)	0.67	0.61	0.72	0.59	0.73	0.57	0.75	0.78	0.72	0.82	0.71	0.83	0.69	0.85	0.55	0.49	0.61	0.48	0.63	0.46	0.65
	Central Period (10–19)	0.67	0.61	0.72	0.59	0.73	0.57	0.75	0.78	0.72	0.82	0.71	0.83	0.69	0.85	0.67	0.61	0.72	0.59	0.73	0.57	0.75
	Final Period (≥20)	0.78	0.72	0.82	0.71	0.83	0.69	0.85	0.78	0.72	0.82	0.71	0.83	0.69	0.85	0.89	0.84	0.92	0.83	0.93	0.81	0.93
	Initial Period (0–4)	0.67	0.61	0.72	0.59	0.73	0.57	0.75	0.67	0.61	0.72	0.59	0.73	0.57	0.75	0.67	0.61	0.72	0.59	0.73	0.57	0.75
	Central Period (5–9)	0.67	0.61	0.72	0.59	0.73	0.57	0.75	0.78	0.72	0.82	0.71	0.83	0.69	0.85	0.67	0.61	0.72	0.59	0.73	0.57	0.75
	Final Period (≥10)	0.78	0.72	0.82	0.71	0.83	0.69	0.85	0.89	0.84	0.92	0.83	0.93	0.81	0.93	0.89	0.84	0.92	0.83	0.93	0.81	0.93
Variable IV	Score Difference (SD)	0.78	0.72	0.82	0.71	0.83	0.69	0.85	0.78	0.72	0.82	0.71	0.83	0.69	0.85	-	-	-	-	-	-	-
Categories	Low 0–2	0.67	0.61	0.72	0.59	0.73	0.57	0.75	0.78	0.72	0.82	0.71	0.83	0.69	0.85	0.67	0.61	0.72	0.59	0.73	0.57	0.75
	Mean 3–5	0.67	0.61	0.72	0.59	0.73	0.57	0.75	0.78	0.72	0.82	0.71	0.83	0.69	0.85	0.67	0.61	0.72	0.59	0.73	0.57	0.75
	High +5	0.67	0.61	0.72	0.59	0.73	0.57	0.75	0.78	0.72	0.82	0.71	0.83	0.69	0.85	0.67	0.61	0.72	0.59	0.73	0.57	0.75
	Low 0–2-(5th)	0.67	0.61	0.72	0.59	0.73	0.57	0.75	0.78	0.72	0.82	0.71	0.83	0.69	0.85	0.67	0.61	0.72	0.59	0.73	0.57	0.75
	Mean 3–5-(5th)	0.67	0.61	0.72	0.59	0.73	0.57	0.75	0.78	0.72	0.82	0.71	0.83	0.69	0.85	0.67	0.61	0.72	0.59	0.73	0.57	0.75

**Table 4 sports-10-00156-t004:** Importance descriptive statistics of the categories.

Contextual Variable	Category	Median	Mean	SD
I. Home advantage	Home	7.50	7.60	±2.16
	Away	8.00	7.65	±2.16
II. Opposition Level	Low Level−	7.00	7.10	±2.67
	Low Level+	7.50	7.00	±2.79
	Mid Level−	8.00	7.65	±2.46
	Low Level+	8.00	7.60	±2.50
	High Level	9.50	8.20	±2.42
III. Set Period	Initial Period (0–9)	6.50 ^A^*^,B^**	6.95	±2.37
	Central Period (10–19)	7.50 ^A^*^,B^*	7.60	±2.16
	Final Period (≥20)	10.00	9.05	±1.54
	Initial Period (0–4)	7.50 ^A^*^,B^*	7.35	±2.23
	Central Period (5–9)	8.00 ^A^*^,B^*	7.60	±2.21
	Final Period (≥10)	10.00	9.20	±1.44
IV. Score Difference	Low 0–2	9.00	7.55	±2.74
	Mean 3–5	7.50	7.45	±2.31
	High +5	8.00	7.35	±2.81
	Low 0–2-(5th)	8.50	7.65	±2.43
	Mean 3–5-(5th)	8.00	7.85	±1.93
	High +5-(5th)	8.00	7.40	±2.78
V. Trend	Winning	7.00	7.30	±2.27
	Losing	7.50	7.55	±2.24
	Tied	8.00	7.90	±2.10
VI. Result previous set	Won	7.00	7.35	±2.41
	Lost	8.00	7.45	±2.42
VII. Competitive Load	Attenuated Load	7.00	7.30	±2.36
	High Load	10.00 *	8.80	±2.07

Notes: SD = standard deviation; ^A^ = significant differences with Final Period ≥ 20; ^B^ = significant differences with Final Period ≥ 10; significance = * *p* < 0.05; ** *p* < 0.01.

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
