# Peer review of "In the Opinion of Elite Volleyball Coaches, How Do Contextual Variables Influence Individual Volleyball Performance in Competitions?"

_sports, 2022, doi:10.3390/sports10100156_

Round 1
Reviewer 1 Report
The paper analyse an interesting side of scouting, identifying relevant variables in the coache's opinion, which can affect the player's performance.
The paper present many concepts and results, in a quasi-narrative way. It is necessary that the concepts and the results are presesented more schematically, because the paper is difficult to read.
Also, the English language must be fixed, both in grammar than in readability.
Overall, an interesting work.
Author Response
Revisor #1: In the opinion of elite volleyball coaches, how do contextual variables influence individual volleyball performance in competitions?
The paper analyse an interesting side of scouting, identifying relevant variables in the coache's opinion, which can affect the player's performance.
Response: We want to thank the reviewer commentaries to improve the quality of the manuscript. In addition, according to the checklist evaluation of the reviewer we have tried to improve the introduction including a paragraph with some concepts of contextual variables (lines 64-66).
“In this line, previous literature has shown that variables like home advantage [16], the difference between opposite team levels [17] or the final moments of the set and the match [18] have an influence on the final result”
The paper present many concepts and results, in a quasi-narrative way. It is necessary that the concepts and the results are presented more schematically, because the paper is difficult to read.
Response: Thank you for your suggestion. We have introduced subtitles to show block of results separately, making it easier to understand all the concepts. (Lines 171, 220 and 242)
3.1. Relevance of the contextual variables
3.2. Definition of the contextual variables
3.3. Importance of the contextual variables
Also, the English language must be fixed, both in grammar than in readability.
Response: We have improved the English language using specialist editing services.

Reviewer 2 Report
General comments
This manuscript aims at identifying the opinions of elite coaches, in terms of relevance, definition and importance, on contextual volleyball variables to measure individual performance in competitions. Results are enriched by several relevant comments by the expert judges. Findings are thoroughly discussed in 4. Discussion. The authors conclude that only opposition level, set period, score difference, result of the previous set and competitive load variables were deemed to be relevant by the experts to measure individual player performances in competitions. Overall, authors manage to fulfill sufficiently their aim.
Minor comments
(line 240) … Z… ?
(l487) … citado… ?
(l336) … matches [40].
Author Response
Revisor #2: In the opinion of elite volleyball coaches, how do contextual variables influence individual volleyball performance in competitions?
This manuscript aims at identifying the opinions of elite coaches, in terms of relevance, definition and importance, on contextual volleyball variables to measure individual performance in competitions. Results are enriched by several relevant comments by the expert judges. Findings are thoroughly discussed in 4. Discussion. The authors conclude that only opposition level, set period, score difference, result of the previous set and competitive load variables were deemed to be relevant by the experts to measure individual player performances in competitions. Overall, authors manage to fulfill sufficiently their aim.
Response: We want to thank the reviewer commentaries to improve the quality of the manuscript and his positive review. In addition, according to the checklist evaluation of the reviewer we have tried to improve the English language using specialist editing services.
Minor comments:
(line 240) … Z… ?
Response: (line 254) The reason for using Z is that the Competitive Load variable is a variable with only 2 categories and therefore it was analysed using the test U Mann-Whitney
(line 464)…citado ?
Response: Thank you for the comment. This was a translation error that we have already corrected as:“cited 2022 March 17”
(Line 515) 31.“Raymond B, Izkowicz A, Lebedew M, Dietz J. The value of points in volleyball. Science Untangled. 2020 [cited 2022 March 17 citado 17 marzo 2022]. Available from: https://untan.gl/point-value.html
(Line 326)…matches [40].
Response: (line 354) Thank you for your suggestion. We have corrected the error by removing the author and leaving only the reference number. “matches Zuccolotto et al., [40]”
